# Analysis of the Mediating Role of Psychological Empowerment between Perceived Leader Trust and Employee Work Performance

**DOI:** 10.3390/ijerph19116712

**Published:** 2022-05-31

**Authors:** Xiaoli Liu, Xiaopeng Ren

**Affiliations:** CAS Key Laboratory of Behavioral Science, Institute of Psychology, Chinese Academy of Sciences, Beijing 100101, China; renxp@psych.ac.cn

**Keywords:** psychological empowerment, perceived leader trust, employee work performance, intrinsic work motivation, perceived leader dependence, employee relationship performance

## Abstract

High levels of trust between employees and leaders moderate the relationship between organizational management practices. A collaborative environment encourages employees to have more Psychological Empowerment, which in turn leads to better performance. Based on Intrinsic Work Motivation and Self-Evaluation, this paper uses Perceived Leader Trust as an independent variable, Employee Work Performance as a dependent variable, and introduces Psychological Empowerment to explore the internal mechanism of perceived trust. This paper proposes a total of 28 hypotheses, and 25 hypotheses have been verified. The specific research conclusions are as follows: (1) Perceived Leader Trust has a positive impact on Employee Work Performance. (2) Perceived Leader Trust positively affects employees’ Psychological Empowerment. Perceived Leader Dependence has a significant impact on all dimensions of Psychological Empowerment, but the relationship between Perceived Information Disclosure and Work Meaning is not significant. (3) Psychological Empowerment is positively correlated with Employee Work Performance, in which the four dimensions of Psychological Empowerment are significantly related to Employee Task Performance, while Work Meaning and Autonomy are not significantly related to Employee Relationship Performance. (4) Psychological Empowerment, as the overall perception of employees, plays a partial mediating role between Perceived Leader Trust and Employee Work Performance. This paper verifies the role of Psychological Empowerment between Perceived Leader Trust and Employee Work Performance, and explores the internal mechanism of Perceived Leader Trust from the perspective of employees’ Intrinsic Work Motivation, which promotes the development of organizational management practices.

## 1. Introduction

In recent years, interpersonal trust in organizations has become an increasingly important research topic. Especially with the rapidly changing business environment and increasing global competitive pressure, organizations face increasing uncertainty in the course of doing business [1]. Companies have now realized that human resources are the key to improving organizational competitiveness, and one of the most important challenges managers face is to build organizational trust at all levels by allowing employees to participate in organizational issues [2,3].

In organizational management, there are two types of trust that have an important impact on employees’ attitudes and behaviors, namely upward trust and downward trust [4]. Employees’ perception of their leaders’ trust is an important type of trust that has emerged in recent years, and is considered to be an important prerequisite for arousing employees’ inner perception [5,6]. Trust and Felt trust are two sides of the same coin. The two are independent constructs, and the most fundamental difference is the difference in the subject of action. The subject of trust is the giver of trust, and the subject of perceiving trust is the perceiver of trust. Sometimes the trust given by the truster may not be felt by the trustee. This is because trust and feeling trusted are different attitudes and views of two parties, which will be affected by the personality characteristics of the trust perceiver and organizational factors that affect trust attributes [7].

The fact that employees in an organization perceive superiors to be trusted only affects their behavior when they feel they are trusted. Therefore, in order to improve the relationship between superiors and subordinates in the organization, it is very important to ensure that employees perceive leader trust [2,5,8]. Trust is the cornerstone of improving organizational effectiveness and reducing employee management costs. Employees who perceive superior trust have higher job satisfaction and organizational commitment, and have better behavior and performance in expressing their opinions at work [9]. Therefore, this study takes the perception side of trust—employee as the starting point, and explores the relationship between Perceived Leader Trust, Psychological Empowerment, and Employee Work Performance from the perspective of Intrinsic Work Motivation and Self-Evaluation. This has important theoretical and practical significance for solving the missing link in perceived trust theory, and provides a reference for organizations to build a high-trust organizational structure.

## 2. Relevant Theoretical Basis

### 2.1. Intrinsic Work Motivation Theory

For individuals, the value of work itself is the intrinsic motivation for work, which is derived from people’s endogenous need for a sense of competence and self-determination [10]. Competence and self-determination are core components of Intrinsic Work Motivation [11]. Amabile proposed a five-factor model of intrinsic motivation, including competency, self-determination, interest, curiosity, and work engagement [12]. Existing studies have shown that the role of employees’ Intrinsic Work Motivation depends not only on individual differences, but also on the influence of the work environment. Therefore, Intrinsic Work Motivation has individual and social characteristics [13]. Deci found that external factors such as trust, mentoring, and participation opportunities from superiors can enhance employees’ Intrinsic Work Motivation [14]. When managers encourage employees, acknowledge their views, trust them, and provide them with guidance and choice at work, employees’ Intrinsic Work Motivation is significantly enhanced, and they tend to perform better at work [15].

### 2.2. Self-Evaluation Theory

Self-evaluation, as the intrinsic motivation of employees, represents how employees evaluate themselves [16], and can have a positive and significant impact on employees’ behavior, resulting in the satisfaction of leaders [17]. Nerstad found that Perceived Leader Trust, as an employee’s internal self-cognition, can enhance employees’ Self-Evaluation and make employees more recognized for their work meaning, ability, and influence [18]. When employees perceive that their leaders trust them, self-assessments related to perceived trust can lead to a stronger sense of competence and autonomy in employees, increase their influence at work, and promote better performance [19,20]. From these findings, we can infer that superior trust in social settings is important for the Self-Evaluation process in the workplace.

### 2.3. Perceived Leader Trust

Employees who perceive higher-level trust have higher accountability norms, organizational commitment, and organizational self-esteem. The positive relationship between employees’ social responsibility and their own behavior becomes stronger when leaders motivate frontline employees to serve customers. Furthermore, when frontline employees are satisfied with their jobs, the relationship between responsibility, self-esteem, and their own behavior is strengthened [21,22]. Lau [23] found that teachers who perceived the principal’s trust had higher task performance and organizational citizenship behavior, and organizational self-esteem played a positive moderating role between perceived trust and organizational citizenship behavior and task performance. Liu Y found that Perceived Leader Trust, as an employee’s individual cognition, has a positive impact on employees’ knowledge sharing behavior and voice behavior [24]. Ma E [25] found that self-efficacy and psychological safety play a dual mediating role between Perceived Leader Trust and voice behavior [26]. Thongpapanl N T and M Leppäniemi found that normative commitment and affective commitment play a mediating role between Perceived Leader Trust and project performance, and the mediating effect of continuous commitment has not been verified [27,28].

### 2.4. Psychological Empowerment

The effects of Psychological Empowerment on employee attitudes and behaviors include job satisfaction, organizational commitment, job engagement, and turnover intention [29]. Khany explored the impact of trust and Psychological Empowerment on teacher job satisfaction. Employees with a high perception of Psychological Empowerment have higher job satisfaction [30]. The four dimensions of Psychological Empowerment play a mediating role between work-related outcomes and employees’ perceptions of their direct leaders [31,32]. Lv M [33] explored the effect of Psychological Empowerment on the relationship between trust in business organizations and employee engagement, and found that Psychological Empowerment had a moderating effect on the relationship between organizational trust and employee engagement. Employees with low perceptions of Psychological Empowerment have a stronger positive relationship between organizational trust and engagement [34]. Sun Y L [35] found that job meaning and self-determination in Psychological Empowerment have a positive impact on employee satisfaction and organizational commitment, work meaning has a negative impact on turnover intention, and self-efficacy has a positive impact on organizational commitment.

### 2.5. Employee Work Performance

The individual factors that affect the Employee Work Performance mainly include personality traits, experience, risk preference and so on. Early research on Employee Work Performance focused on the individual factors of employees. Goldsmith P D found that employees’ work experience has a significant impact on job performance. Employees with higher work experience tend to have stronger competencies at work, and therefore tend to have better performance [36]. Cadsby [37] found that risk takers are more inclined to take aggressive measures, and their performance improvement tends to be higher. Hastings R P explored the impact of personality traits on Employee Work Performance. The extraversion personality trait has a more significant effect on relationship performance, and the open personality trait has no significant effect on Employee Work Performance [38].

The organizational factors that affect Employee Work Performance mainly involve the leadership style, organizational culture, and atmosphere of their leaders [39,40]. Wang P found that proper job design can affect job autonomy, integrity, and job feedback, thereby improving Employee Work Performance [41]. Cheng C explored the impact of organizational support on Employee Work Performance based on social exchange theory. Organizational support can positively affect Employee Work Performance, and job well-being plays a moderating role between the two [42]. Farndale’s study found that employees who Perceived Leader Trust had better performance in the organization, and had a positive impact on Employee Work Performance through emotional commitment [43].

## 3. Research Design and Data Analysis

### 3.1. Measurement of Research Variables

#### 3.1.1. Measurement of Perceived Leader Trust

The more employees perceive leaders, the higher the perception of leaders’ dependence and information disclosure [44]. Lau and Wang Hongli [45] evaluated employees’ trust from two aspects: perceived superior dependence and perceived information disclosure, and the consistency coefficient was 0.916. Gabriel compiled a questionnaire for employees’ trust in their superiors based on the behavioral trust inventory developed by Gillespie, and the internal consistency coefficients of “reliance” and “disclosure” were 0.844, respectively [46]. Drawing lessons from the references [47] in exploring the measurement scales of employees’ perceived trust, this paper divides Perceived Leader Trust into Perceived Leader Dependence and Perceived Information Disclosure, and forms a measurement questionnaire including 2 dimensions and 9 items. The specific items are shown in Table 1.

#### 3.1.2. Measurement of Psychological Empowerment

Spreitzer [48] compiled a research questionnaire with 12 items, and tested the reliability and validity of psychological empowerment. The results showed that the internal consistency of the four dimensions of psychological empowerment was between 0.8 and 0.85. Chenji, K [49] used the questionnaire developed by Spreitzer to measure psychological empowerment in the Chinese context. Avolio, B.J. [50], Wang, G. [25], Stander, M.W. [51], Reza, Khany [52] and many other scholars have confirmed the consistency and validity of the questionnaire. Drawing on the reference [53], this paper divides Psychological Empowerment into Work Meaning, Ability, Autonomy, and Influence, and forms a measurement questionnaire including 4 dimensions and 12 items. The specific items are shown in Table 2.

#### 3.1.3. Measurement of Employee Work Performance

An employee’s work experience has a significant impact on work performance. Employees with higher work experience tend to have stronger competencies at work, and therefore tend to have better performance. Drawing on Han Y [54], Hosamane [55], Kara A [56], Gamage B N [57], Yoestini [58], and Idewele I’s [59] improved Employee Work Performance scale, this paper divides Employee Work Performance into Task Performance and Relationship Performance, including 2 dimensions and 11 items. The specific items are shown in Table 3.

### 3.2. Research Hypothesis

#### 3.2.1. Relationship between Perceived Leader Trust and Employee Work Performance

Based on Intrinsic Work Motivation theory, positive beliefs and high expectations drive employees to perform better [60]. Perceived Leader Trust means that employees perceive the leader’s dependence and information disclosure. The leaders rely more on the knowledge, skills, and judgment of their subordinates when making decisions, which will lead to a strong sense of identity and autonomy for employees, which in turn enables employees to have better performance at work [61,62,63]. When employees feel trusted by their leaders, they get information and clues about work requirements, task completion, and compliance with organizational norms. Workers perform better when they perceive information accurately, receive information, and are willing to respond to information [64]. Perceived Leader Trust may make employees feel better about their organizational members, which can motivate employees to take more responsibility for their work, which in turn improves Employee Work Performance [65]. Therefore, we believe that employees who perceive the trust of their leaders will have better performance. Therefore, the following hypotheses are made:

**Hypothesis** **1** **(H1).**
*Perceived Leader Trust positively affects Employee Work Performance.*


**Hypothesis** **1a** **(H1a).**
*Perceived Leader Dependence positively affects Employee Task Performance.*


**Hypothesis** **1b** **(H1b).**
*Perceived Information Disclosure positively affects Employee Task Performance.*


**Hypothesis** **1c** **(H1c).**
*Perceived Leader Dependence positively affects Employee Relationship Performance.*


**Hypothesis** **1d** **(H1d).**
*Perceived Information Disclosure positively affects Employee Relationship Performance.*


#### 3.2.2. Relationship between Perceived Leader Trust and Psychological Empowerment

Overall Psychological Empowerment increases when employees’ trust in their leaders’ reliability, dependence, and competence increases [66]. Based on the theory of Intrinsic Work Motivation, when employees perceive the dependence and information disclosure of their leaders, their sense of competence, autonomy, and belongingness will be enhanced, and they will have a higher sense of identity with their work, thereby enhancing their overall Psychological Empowerment level [67,68]. When employees perceive positive evaluation and support from their leaders, they attribute it to their self-concept, which significantly affects their Psychological Empowerment perception. Trust improves the relationship and increases the leader’s willingness to delegate authority to subordinates [69]. Managers’ experience of Psychological Empowerment is related to their trust in organizational leaders, and the success of Psychological Empowerment depends on the trust between employees and managers. Employee’s behavior and decision-making are more influenced by leadership or environment [67,70,71]. The trust of leaders is the external environmental factor of employee behavior. Perceiving leaders’ trust will obviously increase the internal Psychological Empowerment of employees [72]. Therefore, we argue that the higher the level of trust employees perceive from their leaders, the stronger their perception of Psychological Empowerment. Therefore, the following hypotheses are made:

**Hypothesis** **2** **(H2).**
*Perceived Leader Trust positively affects Employee Psychological Empowerment.*


**Hypothesis** **2a** **(H2a).**
*Perceived Leader Dependence positively affects Employee Work Meaning.*


**Hypothesis** **2b** **(H2b).**
*Perceived Leader Dependence positively affects Employee Ability.*


**Hypothesis** **2c** **(H2c).**
*Perceived Leader Dependence positively affects Employee Autonomy.*


**Hypothesis** **2d** **(H2d).**
*Perceived Leader Dependence positively affects Employee Influence.*


**Hypothesis** **2e** **(H2e).**
*Perceived Information Disclosure positively affects Employee Work Meaning.*


**Hypothesis** **2f** **(H2f).**
*Perceived Information Disclosure positively affects Employee Ability.*


**Hypothesis** **2g** **(H2g).**
*Perceived Information Disclosure positively affects Employee Autonomy.*


**Hypothesis** **2h** **(H2h).**
*Perceived Information Disclosure positively affects Employee Influence.*


#### 3.2.3. Relationship between Psychological Empowerment and Employee Work Performance

In work situations, employees with a high perception of Psychological Empowerment tend to have a stronger sense of competence and autonomy, and pay more attention to the impact and value of the work itself [73,74]. This means that employees with a high perception of Psychological Empowerment will respond autonomously when faced with risks and uncertainties at work, and have more input in their work, which will promote employees to have higher work performance. Psychological Empowerment is related to management effectiveness and Employee Work Performance [75]. Meaning at work can increase employee focus and loyalty to work. Perceived competence can make employees more resilient in the face of difficulties and challenges, and have a higher pursuit of goals and tasks [76]. Employees who are more influential at work can coordinate resources more smoothly when completing tasks, and organize that line of work, resulting in high work performance [76,77]. Therefore, we believe that employees who perceive Psychological Empowerment will have better performance. Therefore, the following hypotheses are made:

**Hypothesis** **3** **(H3).**
*Psychological Empowerment positively affects Employee Work Performance.*


**Hypothesis** **3a** **(H3a).**
*Work Meaning positively affects Employee Task Performance.*


**Hypothesis** **3b** **(H3b).**
*Ability positively affects Employee Task Performance.*


**Hypothesis** **3c** **(H3c).**
*Autonomy positively affects Employee Task Performance.*


**Hypothesis** **3d** **(H3d).**
*Influence positively affects Employee Task Performance.*


**Hypothesis** **3e** **(H3e).**
*Work Meaning positively affects Employee Relationship Performance.*


**Hypothesis** **3f** **(H3f).**
*Ability positively affects Employee Relationship Performance.*


**Hypothesis** **3g** **(H3g).**
*Autonomy positively affects Employee Relationship Performance.*


**Hypothesis** **3h** **(H3h).**
*Influence positively affects Employee Relationship Performance.*


#### 3.2.4. Mediating Role of Psychological Empowerment

Based on the viewpoints of Intrinsic Work Motivation and self-evaluation, when employees perceive the trust of their leaders, they will positively evaluate themselves, thereby improving their sense of competence, autonomy, and belongingness, and enabling employees to generate Psychological Empowerment. To maintain this Psychological Empowerment, employees who perceive the trust of their leaders strive to improve their work performance [78]. Employees who perceive the trust of their leaders will have a sense of responsibility, as well as Psychological Empowerment, which will motivate employees to perform well at work. When employees perceive their own importance at work and have a high level of self-worth experience, employees will increase their motivation to work harder, and Psychological Empowerment just reflects employees’ self-evaluation [79]. Achieving the desired effect of Psychological Empowerment requires an increased level of trust between employees and their superiors. Subordinates who perceive higher-level trust Intrinsic Work Motivation, enhance organizational citizenship, and motivate employees to stay in the organization [78,80]. Perceived Leader Trust is the process of stimulating employees’ Intrinsic Work Motivation [69]. Psychological Empowerment generated by the Intrinsic Work Motivation makes the tasks assigned by the leaders more meaningful to the employees, and the employees will be better at discovering the value of the work and recognize the tasks assigned in the work more [69,71,79]. This paper argues that Perceived Leader Trust helps to increase the Psychological Empowerment of employees, and the increase of Psychological Empowerment will lead to better performance of employees. Therefore, the following hypotheses are made:

**Hypothesis** **4** **(H4).**
*Psychological Empowerment mediates between Perceived Leader Trust and Employee Work Performance.*


**Hypothesis** **4a** **(H4a).**
*Psychological Empowerment mediates between Perceived Leader Dependence and Employee Task Performance.*


**Hypothesis** **4b** **(H4b).**
*Psychological Empowerment mediates between Perceived Leader Dependence and Employee Relationship Performance.*


**Hypothesis** **4c** **(H4c).**
*Psychological Empowerment mediates between Perceived Information Disclosure and Employee Task Performance.*


**Hypothesis** **4d** **(H4d).**
*Psychological Empowerment mediates between Perceived Information Disclosure and Employee Relationship Performance.*


### 3.3. Questionnaire

The questionnaire designed in this paper contains four parts. The first part is the research description of the questionnaire, including the basic information of the participants in the questionnaire, which contains 6 items. The second part is the measurement of Perceived Leader Trust, which contains 9 items in total, including 4 items for Perceived Leader Dependence and 5 items for Perceived Information Disclosure. The third part is the measurement of Psychological Empowerment, which contains 12 items, including 3 items each for Work Meaning, Ability, Autonomy, and Influence. The fourth part is the measurement of Employee Work Performance, which contains 11 items, including 5 items on Employee Task Performance and 6 items on Employee Relationship Performance.

The survey subjects selected in this paper are mainly employees of different enterprises and institutions.

First, this paper selected MBA students who had participated in actual work in enterprises and institutions. They came from different industries and regions, which made the sample highly reliable and rich. A total of 110 questionnaires were distributed and 101 questionnaires were returned.

Second, relying on Internet social platforms—”WeChat” and “QQ” to distribute questionnaires to classmates, friends, etc., participating in the work, a total of 207 questionnaires were distributed and recovered.

The questionnaire was issued from June 2021 to August 2021. A total of 317 questionnaires were distributed and 308 questionnaires were collected. The sample recovery rate was 97.2%. After removing 21 invalid questionnaires, 287 questionnaires were obtained, and the recovery rate of valid questionnaires was 90.5%.

NOTE: All methods were carried out in accordance with relevant international and Chinese guidelines and regulations. All experimental protocols were approved by Institute of Psychology, Chinese Academy of Sciences, and Ethics Committee of CAS. Moreover, the informed consent was obtained from all subjects and their legal guardian(s).

### 3.4. Descriptive Statistical Analysis

The specific information is shown in Table 4. It can be seen from Table 4 that there is little difference between male and female ratios (52.96% vs. 47.04%). In terms of age, 94.08% of employees are under the age of 40. This is because most of the employees in the current corporate environment are around 20–40 years old, so the number of questionnaires is the largest. From the perspective of educational background, the proportion of “undergraduate” is the largest (49.48%), followed by “master” (35.19%), with a higher degree of education. The higher percent of masters is because the sample includes more MBA. From the work distribution, most of the samples are “General Employee” (58.54%). From the perspective of working age distribution, there are more samples of “1–3 years” and “4–6 years”, accounting for 48.08% and 30.31% respectively. Among them, the proportion of “Private Enterprise” is the largest, at 48.43%.

### 3.5. Reliability and Validity Analysis

#### 3.5.1. Reliability Test of Scale

(1) Reliability analysis of Perceived Leader Trust

As shown in Table 5, the CITC values of 9 items in the 2 dimensions of Perceived Leader Trust are all greater than 0.6, indicating that there is a good correlation between the items of the scale. The Cronbach’s α value of Perceived Leader Dependence was 0.853, and the Cronbach’s α value of Perceived Information Disclosure was 0.871, indicating that the reliability of the scale was good. Data on Perceived Leader Trust is of high quality and can be used for further analysis.

(2) Reliability analysis of Psychological Empowerment

As shown in Table 6, the CITC values of 12 items of Psychological Empowerment are all greater than 0.6, indicating that there is a good correlation coefficient between the items of the scale. The Cronbach’s α corresponding to the four dimensions of Psychological Empowerment is all greater than 0.8, indicating a good level of reliability. The data reliability of Psychological Empowerment is of high quality and can be used for further analysis.

(3) Reliability analysis of Employee Work Performance

As shown in Table 7, the CITC values of 11 items in the two dimensions of work performance are all greater than 0.6, indicating that there is a good correlation coefficient between the items of the scale. The Cronbach’s α corresponding to the two dimensions of job performance is greater than 0.8, indicating a good level of reliability. The data reliability of job performance is of high quality and can be used for further analysis.

#### 3.5.2. Validity Test of Scale

(1) Validity analysis of Perceived Leader Trust

The KMO value and Bartlett’s sphericity test results of Perceived Leader Trust are shown in Table 8 (detailed data such as Table A1 and Table A2). The KMO value of Perceived Leader Trust was 0.862, and the Bartlett sphericity test was significant at the 0.000 level. The sample data illustrating the variable of Perceived Leader Trust can be subjected to factor analysis.

(2) Validity analysis of Psychological Empowerment

The KMO value and Bartlett’s sphericity test results of Psychological Empowerment are shown in Table 9 (detailed data such as Table A3 and Table A4). The KMO value for Psychological Empowerment was 0.833, and the Bartlett test of sphericity was significant at the 0.000 level. The sample data illustrating the variables of Psychological Empowerment can be subjected to factor analysis.

(3) Validity analysis of Employee Work Performance

The KMO value and Bartlett’s sphericity test results of Employee Work Performance are shown in Table 10 (detailed data such as Table A5 and Table A6). The KMO value of Employee Work Performance was 0.944, and the Bartlett sphericity test was significant at the 0.000 level. The sample data illustrating the variables of Employee Work Performance can be subjected to factor analysis.

## 4. Regression Analysis

### 4.1. Regression Analysis of Perceived Leader Trust on Employee Work Performance

(1) Regression analysis of Perceived Leader Trust on Employee Task Performance

Table 11 shows the results of regression analysis of Employee Task Performance on each dimension of Perceived Leader Trust. The multiple correlation coefficient between each dimension of Perceived Leader Trust and Employee Task Performance is 0.663, indicating that there is a positive correlation between the variables. The coefficient of determination *R*^2^ = 0.440 indicates that the data explanation degree of Perceived Leader Trust to Employee Task Performance is 44.0%. In the analysis of variance, *F* = 76.950, *Sig.* = 0.000, indicating that the model is highly significant and statistically significant. In addition, the regression coefficient of Perceived Leader Dependence was 0.483, *Sig.* = 0.000, the regression coefficient of Perceived Information Disclosure is 0.617, *Sig.* = 0.000, indicating that Perceived Leader Dependence and Perceived Information Disclosure are significantly positively correlated with Employee Task Performance. Therefore, the research hypotheses H1a and H1b hold.

(2) Regression analysis of Perceived Leader Trust on Employee Relationship Performance

Table 12 shows the regression analysis results of each dimension of Perceived Leader Trust on Employee Relationship Performance. The multiple correlation coefficient between each dimension of Perceived Leader Trust and Employee Relationship Performance is 0.668, indicating that there is a positive correlation between the variables. The coefficient of determination *R*^2^ = 0.446 indicates that the data explanation degree of Perceived Leader Trust on Employee Relationship Performance is 44.6%. In the analysis of variance, *F* = 78.765, *Sig.* = 0.000, indicating that the model is highly significant and statistically significant. In addition, the regression coefficient of Perceived Leader Dependence is 0.428, *Sig.* = 0.000, and the regression coefficient of Perceived Information Disclosure is 0.494, *Sig.* = 0.000, indicating that Perceived Leader Dependence and Perceived Information Disclosure are significantly positively correlated with Employee Relationship Performance. Therefore, the research hypotheses H1c and H1d hold.

(3) Regression analysis of Perceived Leader Trust on Employee Work Performance

The results of the regression analysis of Perceived Leader Trust on Employee Work Performance are shown in Table 13. The multiple correlation coefficient between Perceived Leader Trust and Employee Work Performance was 0.744, indicating a positive correlation between the variables. The coefficient of determination *R*^2^ = 0.554 indicates that the data explanation degree of Perceived Leader Trust to Employee Work Performance is 55.4%. In the analysis of variance, *F* = 244.646, *Sig.* = 0.000, indicating that the model is highly significant and statistically significant. In addition, the regression coefficient of Perceived Leader Trust is 1.123, and *Sig.* = 0.000 indicates that Perceived Leader Trust is significantly positively correlated with Employee Work Performance. Therefore, the research hypothesis H1 holds.

### 4.2. Regression Analysis of Perceived Leader Trust on Psychological Empowerment

(1) Regression analysis of Perceived Leader Trust on Work Meaning

The results of the regression analysis of each dimension of Perceived Leader Trust on Work Meaning are shown in Table 14. The multiple correlation coefficient between each dimension of Perceived Leader Trust and Work Meaning is 0.294, indicating that there is a positive correlation between the variables. The coefficient of determination *R*^2^ = 0.086, *F* = 9.240 and *Sig.* = 0.000 in the analysis of variance, indicating that the model is highly significant and statistically significant. In addition, the regression coefficient of Perceived Leader Dependence is 0.254, *Sig.* = 0.001, indicating that Perceived Leader Dependence is positively correlated with Work Meaning, the regression coefficient of Perceived Information Disclosure is 0.104, and *Sig.* = 0.184, indicating that there is no significant relationship between Perceived Information Disclosure and Work Meaning. Therefore, the research hypothesis H2a holds and H2e does not hold.

(2) Regression analysis of Perceived Leader Trust on Ability

The results of the regression analysis of each dimension of Perceived Leader Trust on Ability are shown in Table 15. The multiple correlation coefficient between each dimension of Perceived Leader Trust and Ability is 0.418, indicating that there is a positive correlation between the variables. The coefficient of determination *R*^2^ = 0.174, indicating that the data explanation degree of Perceived Leader Trust to Ability is 17.4%. In the analysis of variance, *F* = 20.705, *Sig.* = 0.000, indicating that the model is highly significant and statistically significant. In addition, the regression coefficient of Perceived Leader Dependence was 0.223, *Sig.* = 0.003, the regression coefficient of Perceived Information Disclosure is 0.340, *Sig.* = 0.000, indicating that Perceived Leader Dependence and Perceived Information Disclosure are significantly positively correlated with Ability. Therefore, the research hypotheses H2b and H2f hold.

(3) Regression analysis of Perceived Leader Trust on Autonomy

The results of regression analysis on Autonomy of each dimension of Perceived Leader Trust are shown in Table 16. The multiple correlation coefficient of each dimension of Perceived Leader Trust and Autonomy is 0.477, indicating that there is a positive correlation between the variables. The coefficient of determination *R*^2^ = 0.228, indicating that the data explanation degree of Perceived Leader Trust to Autonomy is 22.8%. In the analysis of variance, *F* = 28.864, *Sig.* = 0.000, indicating that the model is highly significant and statistically significant. In addition, the regression coefficient of Perceived Leader Dependence is 0.267, *Sig.* = 0.000; the regression coefficient of Perceived Information Disclosure is 0.363, *Sig.* = 0.000, indicating that Perceived Leader Dependence and Perceived Information Disclosure are significantly positively correlated with Autonomy. Therefore, the research hypotheses H2c and H2g hold.

(4) Regression analysis of Perceived Leader Trust on Influence

The results of the regression analysis of Influence on each dimension of Perceived Leader Trust are shown in Table 17. The multiple correlation coefficient between each dimension of Perceived Leader Trust and Influence is 0.466, indicating that there is a positive correlation between the variables. The coefficient of determination *R*^2^ = 0.217, indicating that the data interpretation of Perceived Leader Trust to Influence is 21.7%. In the analysis of variance, *F* = 27.219, *Sig.* = 0.000, indicating that the model is highly significant and statistically significant. In addition, the regression coefficient of Perceived Leader Dependence is 0.192, *Sig.* = 0.001; the regression coefficient of Perceived Information Disclosure is 0.434, *Sig.* = 0.000, indicating that Perceived Leader Dependence and Perceived Information Disclosure are significantly positively correlated with Influence. Therefore, the research hypotheses H2d and H2h hold.

(5) Regression analysis of Perceived Leader Trust on Psychological Empowerment

The regression analysis results of Perceived Leader Trust on Psychological Empowerment are shown in Table 18. The multiple correlation coefficient between Perceived Leader Trust and Psychological Empowerment was 0.568, indicating a positive correlation between the variables. The coefficient of determination *R*^2^ = 0.323, indicating that the data explanation degree of Perceived Leader Trust to Psychological Empowerment is 32.3%. In the analysis of variance, *F* = 94.010, *Sig.* = 0.000, indicating that the model has high significance and statistical significance. In addition, the regression coefficient of Perceived Leader Trust is 0.545, *Sig.* = 0.000, indicating that Perceived Leader Trust is significantly positively correlated with Psychological Empowerment. Therefore, the research hypothesis H2 holds.

### 4.3. Regression Analysis of Psychological Empowerment on Employee Work Performance

(1) Regression analysis of Psychological Empowerment on Employee Task Performance

Table 19 shows the regression analysis results of each dimension of Psychological Empowerment on Employee Task Performance. The multiple correlation coefficient between each dimension of Psychological Empowerment and Employee Task Performance is 0.800, indicating that there is a positive correlation between the variables. The coefficient of determination *R*^2^ = 0.640, indicating that the data explanation degree of Psychological Empowerment on Employee Task Performance is 64.0%. In the analysis of variance, *F* = 86.284, *Sig.* = 0.000, indicating that the model is highly significant and statistically significant. In addition, the regression coefficient of Work Meaning is 0.199, *Sig.* = 0.001; the regression coefficient of Ability is 0.259, *Sig. =* 0.000; the regression coefficient for Autonomy is 0.506, *Sig.* = 0.000; the regression coefficient of Influence is 0.376, *Sig.* = 0.000, indicating that Work Meaning, Ability, Autonomy, and Influence are positively related to Employee Task Performance. Therefore, the research hypotheses H3a, H3b, H3c, H3d hold.

(2) Regression analysis of Psychological Empowerment on Employee Relationship Performance

Table 20 shows the regression analysis results of each dimension of Psychological Empowerment on Employee Relationship Performance. The multiple correlation coefficient of each dimension of Psychological Empowerment and Employee Relationship Performance is 0.626, indicating that there is a positive correlation between the variables. The coefficient of determination *R*^2^ = 0.392, indicating that the data explanation degree of Psychological Empowerment on Employee Relationship Performance is 39.2%. In the analysis of variance, *F* = 31.332, *Sig.* = 0.000, indicating that the model is highly significant and statistically significant. In addition, the regression coefficient of Work Meaning is 0.115, *Sig.* = 0.080, indicating that there is no significant relationship between Work Meaning and Employee Relationship Performance. The regression coefficient of Ability is 0.294, *Sig.* = 0.000, indicating that Ability has a significant positive correlation with Employee Relationship Performance. The regression coefficient of Autonomy is 0.097, *Sig.* = 0.162, indicating that there is no significant relationship between Autonomy and Employee Relationship Performance. The regression coefficient of Influence is 0.350, *Sig.* = 0.000, indicating that Influence is positively correlated with Employee Relationship Performance. Therefore, the research hypotheses H3f and H3h hold, but H3e and H3g do not hold.

(3) Regression analysis of Psychological Empowerment on Employee Work Performance

The results of the regression analysis of Psychological Empowerment on Employee Work Performance are shown in Table 21. The multiple correlation coefficient between Psychological Empowerment and Employee Work Performance was 0.787, indicating a positive correlation between the variables. The coefficient of determination *R*^2^ = 0.620 indicates that the data explanation degree of Psychological Empowerment to Employee Work Performance is 62.0%. In the analysis of variance, *F* = 320.877, *Sig.* = 0.000, indicating that the model is highly significant and statistically significant. In addition, the regression coefficient of Psychological Empowerment is 1.240, *Sig.* = 0.000, indicating that Psychological Empowerment is significantly positively correlated with Employee Work Performance. Therefore, the research hypothesis H3 holds.

## 5. Analysis of the Mediating Effect of Psychological Empowerment

In this paper, the independent variable is ***X***, the dependent variable is ***Y***, and the mediating variable is ***M*** to construct a Psychological Empowerment Mediation Model. The specific situation is shown in Figure 1.

Step 1: Exploring whether the regression analysis coefficient ***a*** of ***X*** to ***M*** is significant. If it is not significant, stop the mediation test; if it is significant, go to Step 2;

Step 2: Explore whether the regression analysis coefficient ***b*** of ***Y*** to ***M*** is significant. If it is not significant, stop the mediation test; if it is significant, go to Step 3;

Step 3: Introducing ***X*** and ***M*** into the regression equation to explore the combined effect of ***X*** and ***M*** on ***Y***. If the regression coefficient ***c*** of ***X*** and the regression coefficient ***d*** of ***M*** are both significant, it means that ***M*** plays a partial mediating role. If the regression coefficient ***c*** of ***X*** is significant, the regression coefficient ***d*** of ***M*** is not significant, indicating that ***M*** plays a complete mediating role. If the regression coefficient ***c*** of ***X*** is not significant, a Soble test is required. If the Soble test is significant, it means that ***M*** plays a partial mediating role; if the Soble test is not significant, it means that there is no mediating effect.

### 5.1. Mediating Role of Psychological Empowerment between Perceived Leader Dependence and Employee Task Performance

As shown in Table 22:

Model 1: Regression analysis of Perceived Leader Dependence on Employee Task Performance. *R*^2^ = 0.263, *F* = 70.356, *Sig.* = 0.000, the regression coefficient of Perceived Leader Trust is 0.679 (*Sig.* = 0.000), and the regression effect is significant.

Model 2: Regression analysis of Psychological Empowerment on Employee Task Performance. *R*^2^ = 0.616, *F* = 316.535, *Sig.* = 0.000, the regression coefficient of Psychological Empowerment is 1.360 (*Sig.* = 0.000), and the regression effect is significant.

Model 3: Regression analysis of Perceived Leader Dependence and Psychological Empowerment on Employee Task Performance. *R*^2^ = 0.653, *F* = 184.093, *Sig.* = 0.000, the regression coefficient of Psychological Empowerment is 1.200, the regression coefficient of Perceived Leader Dependence is 0.280, and the significance level has not changed.

The above results suggest that Psychological Empowerment plays a partial mediating role between Perceived Leader Dependence and Employee Task Performance, so hypothesis H4a holds.

### 5.2. Mediating Role of Psychological Empowerment between Perceived Leader Dependence and Employee Relationship Performance

As shown in Table 23:

Model 1: Regression analysis of Perceived Leader Dependence on Employee Relationship Performance. *R*^2^ = 0.282, *F* = 77.513, *Sig.* = 0.000, the regression coefficient of Perceived Leader Dependence is 0.586 (*Sig.* = 0.000), and the regression effect is significant.

Model 2: Regression analysis of Psychological Empowerment on Employee Relationship Performance. *R*^2^ = 0.366, *F* = 113.832, *Sig*. = 0.000, the regression coefficient of Psychological Empowerment was 0.873 (*Sig.* = 0.000), and the regression effect was significant.

Model 3: Regression analysis of Perceived Leader Dependence and Psychological Empowerment on Employee Relationship Performance. *R*^2^ = 0.455, *F* = 81.764, *Sig.* = 0.000, the regression coefficient of Psychological Empowerment is 0.665, the regression coefficient of Perceived Leader Dependence is 0.365, and the significance level has not changed.

The above results indicate that Psychological Empowerment plays a partial mediating role between Perceived Leader Dependence and Employee Relationship Performance, so hypothesis H4b holds.

### 5.3. Mediating Role of Psychological Empowerment between Perceived Information Disclosure and Employee Task Performance

As shown in Table 24:

Model 1: Regression analysis of Perceived Information Disclosure on Employee Task Performance. *R*^2^ = 0.322, ***F*** = 93.482, *Sig.* = 0.000, the regression coefficient of Perceived Leader Trust is 0.786 (*Sig.* = 0.000), and the regression effect is significant.

Model 2: Regression analysis of Psychological Empowerment on Employee Task Performance. *R*^2^ = 0.616, *F* = 316.535, *Sig.* = 0.000, the regression coefficient of Psychological Empowerment is 1.360 (*Sig.* = 0.000), and the regression effect is significant.

Model 3: Regression analysis of Perceived Information Disclosure and Psychological Empowerment on Employee Task Performance. *R*^2^ = 0.660, *F* = 190.472, *Sig.* = 0.000, the regression coefficient of Psychological Empowerment is 1.156, the regression coefficient of Perceived Information Disclosure is 0.333, and the significance level has not changed.

The above results indicate that Psychological Empowerment plays a partial mediating role between Perceived Information Disclosure and Employee Task Performance, so hypothesis H4c holds.

### 5.4. Mediating Role of Psychological Empowerment between Perceived Information Disclosure and Employee Relationship Performance

As shown in Table 25:

Model 1: Regression analysis of Perceived Information Disclosure on Employee Relationship Performance. *R*^2^ = 0.311, *F* = 89.085, *Sig.* = 0.000, the regression coefficient of Perceived Leader Trust is 0.643 (*Sig.* = 0.000), and the regression effect is significant.

Model 2: Regression analysis of Psychological Empowerment on Employee Relationship Performance. *R*^2^ = 0.366, *F* = 113.832, *Sig.* = 0.000, the regression coefficient of Psychological Empowerment was 0.873 (*Sig.* = 0.000), and the regression effect was significant.

Model 3: Regression analysis of Perceived Information Disclosure and Psychological Empowerment on Employee Relationship Performance. *R*^2^ = 0.456, *F* = 82.222, *Sig.* = 0.000, the regression coefficient of Psychological Empowerment is 0.629, the regression coefficient of Perceived Information Disclosure is 0.397, and the significance level has not changed.

The above results indicate that Psychological Empowerment plays a partial mediating role between Perceived Information Disclosure and Employee Relationship Performance, so hypothesis H4d holds.

### 5.5. Mediating Role of Psychological Empowerment between Perceived Leader Trust and Employee Work Performance

As shown in Table 26:

Model 1: Regression analysis of Perceived Leader Trust on Employee Work Performance. *R*^2^ = 0.554, *F* = 244.646, *Sig.* = 0.000, the regression coefficient of Perceived Leader Trust is 1.123 (*Sig.* = 0.000), and the regression effect is significant.

Model 2: Regression analysis of Psychological Empowerment on Employee Work Performance. *R*^2^ = 0.620, *F* = 320.877, *Sig.* = 0.000, the regression coefficient of Psychological Empowerment is 1.240 (*Sig.* = 0.000), and the regression effect is significant.

Model 3: Regression analysis of Perceived Leader Trust and Psychological Empowerment on Employee Work Performance. *R*^2^ = 0.750, *F* = 293.684, *Sig.* = 0.000, the regression coefficient of Psychological Empowerment is 0.847, the regression coefficient of Perceived Leader Trust is 0.662, and the significance level has not changed.

The above results indicate that Psychological Empowerment plays a partial mediating role between Perceived Leader Trust and Employee Work Performance, so hypothesis H4 holds.

### 5.6. Test Results of Research Hypotheses

A total of 28 research hypotheses are proposed in this paper, of which 25 research hypotheses are valid and 3 research hypotheses are not valid. The specific results are shown in Table 27.

The verification results show that the research hypotheses H2e, H3e, and H3g do not hold. The specific situation is as follows: Perceived Information Disclosure is not related to Employee Work Meaning; Employee Work Meaning and Employee Autonomy are not related to Employee Relationship Performance. In addition, Psychological Empowerment, as an overall mediating variable, played a partial mediating role in the testing of all mediating effects.

## 6. Discussion

(1) Perceived Leader Trust positively affects Employee Work Performance

The regression coefficients of Perceived Leader Dependence and Perceived Information Disclosure on Employee Task Performance are 0.483 and 0.617; the regression coefficients of Perceived Leader Dependence and Perceived Information Disclosure on Employee Relationship Performance are 0.428 and 0.494. This result shows that employees will more actively complete the tasks assigned by their leaders because they perceive their leaders’ dependence and information disclosure, and thus have better performance at work.

(2) Perceived Leader Trust positively affects employees’ Psychological Empowerment level

The empirical results show that Perceived Leader Trust has a positive impact on the overall Psychological Empowerment of employees. Perceived Leader Dependence has a significant positive effect on the Work Meaning, Ability, Autonomy and Influence of Psychological Empowerment, and Perceived Information Disclosure has a positive impact on employees’ Ability, Autonomy, and Influence. Perceived trust based on dependence and information disclosure is built on the emotional connection, interpersonal interest, and support of leaders and subordinates, while employees’ perception of Psychological Empowerment is closely linked to superiors’ communication and support. Therefore, Perceived Leader Trust can positively affect employees’ Psychological Empowerment.

(3) Psychological Empowerment positively affects Employee Work Performance

The empirical results show that employees’ overall Psychological Empowerment has a positive impact on Employee Work Performance. Employees with high Psychological Empowerment tend to be proactive in their work, and have more input in their work, which in turn promotes employees to have higher Employee Work Performance. The four dimensions of Psychological Empowerment can positively affect Employee Task Performance, the Ability and Influence of Psychological Empowerment have a positive impact on Employee Relationship Performance, and Work Meaning and Autonomy have no significant impact on Employee Relationship Performance. The reason for this result is that Employee Relationship Performance is more dependent on the performance and influence of employees at work. However, employees’ perception of Autonomy emphasizes the degree of employees’ self-determination of work, which is not much related to Employee Relationship Performance.

(4) Psychological Empowerment plays a partial mediating role between Perceived Leader Trust and Employee Work Performance

Psychological Empowerment, as a whole, plays a partial mediating role between Perceived Leader Dependence and Employee Task Performance, and partially mediates between Perceived Leader Dependence and Employee Relationship Performance. When employees feel the trust of their leaders, their Employee Work Performance is positively affected, and the effect of Perceived Leader Trust can be explained by changes in employees’ Psychological Empowerment.

## 7. Conclusions

Based on the research results of previous scholars, this paper constructs a theoretical model of Perceived Leader Trust, Psychological Empowerment and Employee Work Performance, and proposes 28 research hypotheses. Among them, Perceived Leader Trust is divided into Perceived Leader Dependence and Perceived Information Disclosure; Psychological Empowerment is divided into Work Meaning, Ability, Autonomy, and Influence; Employee Work Performance is divided into Employee Task Performance and Employee Relationship Performance. This paper adopts a combination of online (WeChat and QQ) and offline (MBA students) methods to collect 308 research data, verify the theoretical model and research hypothesis constructed in this paper through empirical analysis, and finally draw the research conclusion.

(1) Perceived Leader Trust has a positive impact on Employee Work Performance. (2) Perceived Leader Trust can positively affect employees’ perception of Psychological Empowerment. Among them, Perceived Leader Dependence has a significant impact on all dimensions of Psychological Empowerment, but the relationship between Perceived Information Disclosure and Work Meaning is not significant. (3) Employees’ Psychological Empowerment perception is positively related to their work performance. Among them, the four dimensions of Psychological Empowerment are significantly related to Employee Task Performance, and the relationship between Work Meaning and Autonomy and Employee Relationship Performance is not significant. (4) Psychological Empowerment, as the overall perception of employees, plays a partial mediating role between Perceived Leader Trust and Employee Work Performance.

## Figures and Tables

**Figure 1 ijerph-19-06712-f001:**
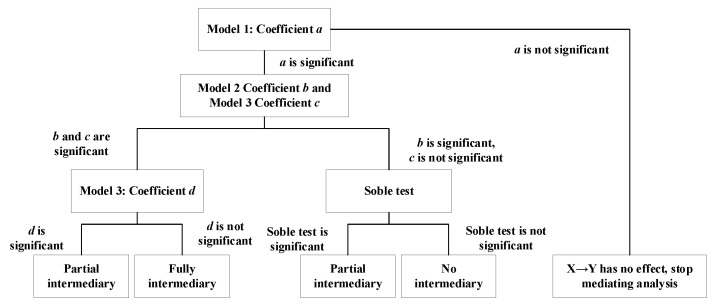
Construction of Psychological Empowerment Mediation Model.

**Table 1 ijerph-19-06712-t001:** Perceived Leader Trust Measurement Items.

Dimension	No.	Items
Perceived Leader Dependence	A11	My direct leader is willing to put me in charge of projects that are important to him
	A12	My direct leader won’t worry about me doing things against him at work
	A13	What my direct leader thinks is important, he will try to get me involved and have an impact
	A14	My direct leader would be more than willing to entrust me with key tasks
Perceived Information Disclosure	A21	My direct leader is willing to share his experience on the job with me
	A22	My direct leader is willing to tell me about mistakes he made at work
	A23	My direct leader is willing to share his views on some sensitive issues with me
	A24	When I have doubts at work, my direct leader will tell me the details of the problem without reservation
	A25	My direct leader is willing to share personal life or family-related information with me

**Table 2 ijerph-19-06712-t002:** Psychological Empowerment Measurement Items.

Dimension	No.	Items
Work Meaning	B11	My work is very meaningful to me
	B12	What I do at work means a lot to me personally
	B13	My work is very important to me
Ability	B21	I can decide for myself how my work is done
	B22	I can decide for myself how to do the work given to me
	B23	At work, I have a lot of autonomy and independence
Autonomy	B31	I have all the skills I need to get the job done
	B32	I am confident that I have all the abilities to do a good job
	B33	I am confident in my ability to get the job done
Influence	B41	I have a greater facilitation of what happens in the department
	B42	I have greater control over what happens in the department
	B43	I have a greater influence on what happens in the department

**Table 3 ijerph-19-06712-t003:** Employee Work Performance Measurement Items.

Dimension	No.	Items
Employee Task Performance	C11	I rarely make the same mistakes at work
	C12	My work always meets the standards required by leader
	C13	I often plan work and advance work
	C14	My work is always productive and on time
	C15	My work performance is quite outstanding in the company
Employee Relationship Performance	C21	I work well with colleagues in a work team
	C22	I provide support and encouragement when colleagues have problems
	C23	I am often enthusiastic and proactive in solving problems at work
	C24	I will often take the initiative to take on additional workloads and strive for better team performance
	C25	When the leader is not present, I still follow his instructions to complete the work
	C26	I expect to be assigned or placed in challenging work

**Table 4 ijerph-19-06712-t004:** Descriptive Statistics of Basic Information.

Name	Option	Frequency	Percentage (%)
Gender	Male	135	47.04
	Female	152	52.96
Age	<25	54	18.82
	26–30	132	45.99
	31–40	84	29.27
	41–50	13	4.53
	>50	4	1.39
Education	High school and below	5	1.74
	Junior College	39	13.59
	Undergraduate	142	49.48
	Master and above	101	35.19
Position	General Employee	168	58.54
	Grassroots manager	66	23.01
	Middle manager	48	16.72
	Senior management	5	1.73
Length of service	<1 year	11	3.83
	1–3 years	138	48.08
	4–6 years	87	30.31
	7–9 years	42	14.64
	>10 years	9	3.14
Unit nature	Private Enterprise	139	48.43
	State-owned enterprise	78	27.18
	Institutions	45	15.68
	Joint venture	21	7.32
	Foreign companies	4	1.39

**Table 5 ijerph-19-06712-t005:** Reliability Analysis of Perceived Leader Trust.

Variable	Item	CITC	α Coefficient	Cronbach’s α
Perceived Leader Dependence	A11	0.699	0.812	0.853
	A12	0.684	0.818	
	A13	0.71	0.807	
	A14	0.689	0.817	
Perceived Information Disclosure	A21	0.651	0.855	0.871
	A22	0.714	0.839	
	A23	0.73	0.835	
	A24	0.717	0.839	
	A25	0.672	0.849	

**Table 6 ijerph-19-06712-t006:** Reliability Analysis of Psychological Empowerment.

Variable	Item	CITC	α Coefficient	Cronbach’s α
Work Meaning	B11	0.701	0.790	0.843
	B12	0.707	0.784	
	B13	0.719	0.772	
Ability	B21	0.741	0.797	0.860
	B22	0.738	0.802	
	B23	0.727	0.811	
Autonomy	B31	0.689	0.753	0.826
	B32	0.695	0.746	
	B33	0.663	0.779	
Influence	B41	0.673	0.789	0.833
	B42	0.726	0.736	
	B43	0.683	0.779	

**Table 7 ijerph-19-06712-t007:** Reliability Analysis of Employee Work Performance.

Variable	Item	CITC	α Coefficient	Cronbach’s α
Employee Task performance	C11	0.864	0.940	0.951
	C12	0.868	0.940	
	C13	0.843	0.944	
	C14	0.887	0.937	
	C15	0.868	0.939	
Employee Relationship performance	C21	0.801	0.908	0.925
	C22	0.782	0.908	
	C23	0.782	0.904	
	C24	0.784	0.908	
	C25	0.787	0.911	
	C26	0.779	0.902	

**Table 8 ijerph-19-06712-t008:** KMO and Bartlett Tests of Perceived Leader Trust.

Sampling adequacy of KMO metrics		0.862
Bartlett sphericity test	Approximate chi-square	812.476
	df	36
	p.	0.000

**Table 9 ijerph-19-06712-t009:** KMO and Bartlett Tests of Psychological Empowerment.

Sampling adequacy of KMO metrics		0.833
Bartlett sphericity test	Approximate chi-square	1109.732
	df	66
	p.	0.000

**Table 10 ijerph-19-06712-t010:** KMO and Bartlett Tests of Employee Work Performance.

Sampling adequacy of KMO metrics		0.944
Bartlett sphericity test	Approximate chi-square	1824.774
	df	45
	p.	0.000

**Table 11 ijerph-19-06712-t011:** Regression Analysis of Perceived Leader Trust on Employee Task Performance.

Model	Model Summary		Variance Analysis		Unstandardized Coefficients		*t*	*Sig.*
	*R*	*R* ^2^	*F*	*Sig.*	*B*	*Std. Error*		
(Constant)					−0.070	0.344	−0.204	0.839
Perceived Leader Dependence	0.663	0.440	76.950	0.000	0.483	0.075	6.426	0.000
Perceived Information Disclosure					0.617	0.079	7.863	0.000

**Table 12 ijerph-19-06712-t012:** Regression Analysis of Perceived Leader Trust on Employee Relationship Performance.

Model	Model Summary		Variance Analysis		Unstandardized Coefficients		*t*	*Sig.*
	*R*	*R* ^2^	*F*	*Sig.*	*B*	*Std. Error*		
(Constant)					0.427	0.285	1.501	0.135
Perceived Leader Dependence	0.668	0.446	78.765	0.000	0.428	0.062	6.888	0.000
Perceived Information Disclosure					0.494	0.065	7.596	0.000

**Table 13 ijerph-19-06712-t013:** Regression Analysis of Perceived Leader Trust on Employee Work Performance.

Model	Model Summary		Variance Analysis		Unstandardized Coefficients		*t*	*Sig.*
	*R*	*R* ^2^	*F*	*Sig.*	*B*	*Std. Error*		
(Constant)					0.199	0.278	0.716	0.475
Perceived Leader Trust	0.744	0.554	244.646	0.000	1.123	0.072	15.641	0.000

**Table 14 ijerph-19-06712-t014:** Regression Analysis of Perceived Leader Trust on Work Meaning.

Model	Model Summary		Variance Analysis		Unstandardized Coefficients		*t*	*Sig.*
	*R*	*R* ^2^	*F*	*Sig.*	*B*	*Std. Error*		
(Constant)					2.592	0.341	7.610	0.000
Perceived Leader Dependence	0.294	0.086	9.240	0.000	0.254	0.074	3.408	0.001
Perceived Information Disclosure					0.104	0.078	1.334	0.184

**Table 15 ijerph-19-06712-t015:** Regression Analysis of Perceived Leader Trust on Ability.

Model	Model Summary		Variance Analysis		Unstandardized Coefficients		*t*	*Sig.*
	*R*	*R* ^2^	*F*	*Sig.*	*B*	*Std. Error*		
(Constant)					1.762	0.341	5.166	0.000
Perceived Leader Dependence	0.418	0.174	20.705	0.000	0.223	0.074	2.996	0.003
Perceived Information Disclosure					0.340	0.078	4.371	0.000

**Table 16 ijerph-19-06712-t016:** Regression Analysis of Perceived Leadership Trust on Autonomy.

Model	Model Summary		Variance Analysis		Unstandardized Coefficients		*t*	*Sig.*
	*R*	*R* ^2^	*F*	*Sig.*	*B*	*Std. Error*		
(Constant)					1.573	0.322	4.887	0.000
Perceived Leader Dependence	0.477	0.228	28.864	0.000	0.267	0.070	3.795	0.000
Perceived Information Disclosure					0.363	0.074	4.941	0.000

**Table 17 ijerph-19-06712-t017:** Regression Analysis of Perceived Leadership Trust on Influence.

Model	Model Summary		Variance Analysis		Unstandardized Coefficients		*t*	*Sig.*
	*R*	*R* ^2^	*F*	*Sig.*	*B*	*Std. Error*		
(Constant)					1.573	0.322	4.887	0.000
Perceived Leader Dependence	0.466	0.217	27.219	0.000	0.192	0.074	2.609	0.001
Perceived Information Disclosure					0.434	0.077	5.637	0.000

**Table 18 ijerph-19-06712-t018:** Regression Analysis of Perceived Leader Trust on Psychological Empowerment.

Model	Model Summary		Variance Analysis		Unstandardized Coefficients		*t*	*Sig.*
	*R*	*R* ^2^	*F*	*Sig.*	*B*	*Std. Error*		
(Constant)					1.852	0.218	8.512	0.000
Perceived Leader Trust	0.568	0.323	94.010	0.000	0.545	0.056	9.696	0.000

**Table 19 ijerph-19-06712-t019:** Regression Analysis of Psychological Empowerment on Employee Task Performance.

Model	Model Summary		Variance Analysis		Unstandardized Coefficients		*t*	*Sig.*
	*R*	*R* ^2^	*F*	*Sig.*	*B*	*Std. Error*		
(Constant)					−1.144	0.299	−3.820	0.000
Work Meaning	0.800	0.640	86.284	0.000	0.199	0.060	3.304	0.001
Ability					0.259	0.062	4.147	0.000
Autonomy					0.506	0.064	7.902	0.000
Influence					0.376	0.059	6.431	0.000

**Table 20 ijerph-19-06712-t020:** Regression Analysis of Psychological Empowerment on Employee Relationship Performance.

Model	Model Summary		Variance Analysis		Unstandardized Coefficients		*t*	*Sig.*
	*R*	*R* ^2^	*F*	*Sig.*	*B*	*Std. Error*		
(Constant)					0.607	0.324	1.875	0.062
Work Meaning	0.626	0.392	31.332	0.000	0.115	0.065	1.761	0.080
Ability					0.294	0.068	4.344	0.000
Autonomy					0.097	0.069	1.403	0.162
Influence					0.350	0.063	5.521	0.000

**Table 21 ijerph-19-06712-t021:** Regression Analysis of Psychological Empowerment on Employee Work Performance.

Model	Model Summary		Variance Analysis		Unstandardized Coefficients		*t*	*Sig.*
	*R*	*R* ^2^	*F*	*Sig.*	*B*	*Std. Error*		
(Constant)					−0.382	0.275	−1.387	0.167
Psychological Empowerment	0.787	0.620	320.877	0.000	1.240	0.069	17.913	0.000

**Table 22 ijerph-19-06712-t022:** Test of the Mediating Effect of Psychological Empowerment between Perceived Leader Dependence and Employee Task Performance.

Regression Model			Variance Analysis		Coefficient Analysis	
X = Perceived Leader Dependence; M = Psychological Empowerment; Y = Employee Task Performance		*R* ^2^	*F*	*Sig.*	*B*	*Sig.*
Model 1: X → Y		0.263	70.356	0.000	0.679	0.000
Model 2: M → Y		0.616	316.535	0.000	1.360	0.000
Model 3: X and M → Y	M → Y	0.653	184.093	0.000	1.200	0.000
	X → Y				0.280	0.000
Conclusion	Psychological Empowerment plays a partial mediating role between Perceived Leader Dependence on Employee Task Performance.					

**Table 23 ijerph-19-06712-t023:** Test of the Mediating Effect of Psychological Empowerment between Perceived Leader Dependence and Employee Relationship Performance.

Regression Model			Variance Analysis		Coefficient Analysis	
X = Perceived Leader Dependence; M = Psychological Empowerment; Y = Employee Relationship Performance		*R* ^2^	*F*	*Sig.*	*B*	*Sig.*
Model 1: X → Y		0.282	77.513	0.000	0.586	0.000
Model 2: M → Y		0.366	113.832	0.000	0.873	0.000
Model 3: X and M → Y	M → Y	0.455	81.764	0.000	0.665	0.000
	X → Y				0.365	0.000
Conclusion	Psychological Empowerment plays a partial mediating role between Perceived Leader Dependence and Employee Relationship Performance.					

**Table 24 ijerph-19-06712-t024:** Test of the Mediating Effect of Psychological Empowerment between Perceived Information Disclosure and Employee Task Performance.

Regression Model			Variance Analysis		Coefficient Analysis	
X = Perceived Leader Dependence; M = Psychological Empowerment; Y = Employee Task Performance		*R* ^2^	*F*	*Sig.*	*B*	*Sig.*
Model 1: X → Y		0.322	93.482	0.000	0.786	0.000
Model 2: M → Y		0.616	316.535	0.000	1.360	0.000
Model 3: X and M → Y	M → Y	0.660	190.472	0.000	1.156	0.000
	X → Y				0.333	0.000
Conclusion	Psychological Empowerment plays a partial mediating role between Perceived Information Disclosure and Employee Task Performance.					

**Table 25 ijerph-19-06712-t025:** Test of the Mediating Effect of Psychological Empowerment between Perceived Information Disclosure and Employee Relationship Performance.

Regression Model			Variance Analysis		Coefficient Analysis	
X = Perceived Information Disclosure; M = Psychological Empowerment; Y = Employee Relationship Performance		*R* ^2^	*F*	*Sig.*	*B*	*Sig.*
Model 1: X → Y		0.311	89.085	0.000	0.643	0.000
Model 2: M → Y		0.366	113.832	0.000	0.873	0.000
Model 3: X and M → Y	M → Y	0.456	82.222	0.000	0.629	0.000
	X → Y				0.397	0.000
Conclusion	Psychological Empowerment plays a partial mediating role between Perceived Information Disclosure and Employee Relationship Performance.					

**Table 26 ijerph-19-06712-t026:** Test of the Mediating Effect of Psychological Empowerment between Perceived Leader Trust and Employee Work Performance.

Regression Model			Variance Analysis		Coefficient Analysis	
X = Perceived Leader Trust; M = Psychological Empowerment; Y = Employee Work Performance		*R* ^2^	*F*	*Sig.*	*B*	*Sig.*
Model 1: X → Y		0.544	244.646	0.000	1.123	0.000
Model 2: M → Y		0.620	320.877	0.000	1.240	0.000
Model 3: X and M → Y	M → Y	0.750	293.684	0.000	0.847	0.000
	X → Y				0.662	0.000
Conclusion	Psychological Empowerment plays a partial mediating role between Perceived Leader Trust and Employee Work Performance.					

**Table 27 ijerph-19-06712-t027:** Summary of Research Hypotheses.

No.	Research Hypotheses	Test Result
H1	Perceived Leader Trust positively affects Employee Work Performance.	Valid
H1a	Perceived Leader Dependence positively affects Employee Task Performance.	Valid
H1b	Perceived Information Disclosure positively affects Employee Task Performance.	Valid
H1c	Perceived Leader Dependence positively affects Employee Relationship Performance.	Valid
H1d	Perceived Information Disclosure positively affects Employee Relationship Performance.	Valid
H2	Perceived Leader Trust positively affects Employee Psychological Empowerment.	Partial Valid
H2a	Perceived Leader Dependence positively affects Employee Work Meaning.	Valid
H2b	Perceived Leader Dependence positively affects Employee Ability.	Valid
H2c	Perceived Leader Dependence positively affects Employee Autonomy.	Valid
H2d	Perceived Leader Dependence positively affects Employee Influence.	Valid
H2e	Perceived Information Disclosure positively affects Employee Work Meaning.	Non-valid
H2f	Perceived Information Disclosure positively affects Employee Ability.	Valid
H2g	Perceived Information Disclosure positively affects Employee Autonomy.	Valid
H2h	Perceived Information Disclosure positively affects Employee Influence.	Valid
H3	Psychological Empowerment positively affects Employee Work Performance.	Partial Valid
H3a	Work Meaning positively affects Employee Task Performance.	Valid
H3b	Ability positively affects Employee Task Performance.	Valid
H3c	Autonomy positively affects Employee Task Performance.	Valid
H3d	Influence positively affects Employee Task Performance.	Valid
H3e	Work Meaning positively affects Employee Relationship Performance.	Non-valid
H3f	Ability positively affects Employee Relationship Performance.	Valid
H3g	Autonomy positively affects Employee Relationship Performance.	Non-valid
H3h	Influence positively affects Employee Relationship Performance.	Valid
H4	Psychological Empowerment mediates between Perceived Leader Trust and Employee Work Performance.	Partial Mediation
H4a	Psychological Empowerment mediates between Perceived Leader Dependence and Employee Task Performance.	Partial Mediation
H4b	Psychological Empowerment mediates between Perceived Leader Dependence and Employee Relationship Performance.	Partial Mediation
H4c	Psychological Empowerment mediates between Perceived Information Disclosure and Employee Task Performance.	Partial Mediation
H4d	Psychological Empowerment mediates between Perceived Information Disclosure and Employee Relationship Performance.	Partial Mediation

## Data Availability

Any data requirements can be obtained by contacting the corresponding author.

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
