# Peer review of "Analysis of the Mediating Role of Psychological Empowerment between Perceived Leader Trust and Employee Work Performance"

_ijerph, 2022, doi:10.3390/ijerph19116712_

Round 1

Reviewer 1 Report

I read the revised version of the manuscript with some degree of fright, as the previous version was problematic. Yet authors managed to solve most of the problems addressed in the previous review, bringing the paper to a new level of clarity. This is highly appreciated. There are still some minor things the authors could work on, to make the paper a high impact on the world of work psychology. I will list some of the improvement suggestions below:

  1. Theoretical introduction:

The authors in the text sentence relating to the reference 9 write about "better voice behavior" of amployess. I do not get this phrase. Could You please explain/rephrase?

Employees who perceive superior trust have higher job satisfaction and organizational commitment, and have better voice behavior and performance at work

2. The reference 17 should relate to the formation of work self-work concept. (The process and results of evaluation have an important impact on the formation of employees' work self-concept[17]). Yet the referenced paper is: J Gómez-Odriozola, Calvete E. Effects of a Mindfulness-based Intervention on Adolescents' Depression and Self-concept: The Moderating Role of Age[J]. Journal of Child and Family Studies, 2021, 30(2): 1501-1515.
I believe this paper does not support the argument the authors bring up.

3. Similarly as in the previous point, I see a problem with the following references:

21, 22; Authors write: "Employees who perceive the trust of their superiors have higher responsibility norms, organizational commitment and organizational self-esteem, and have a higher degree of motivation and completion for the work assigned by their superiors[21,22].", but the references talk about farmers and flood risk, and not the manger - employee relation/trust.

[21]Ambali O I, Begho T. Examining the relationship between farmers' perceived trust and investment preferences[J]. Journal of International Development, 2021, 33(8): 1290-1303.
[22] Zhang K, Parks-Stamm E J, Ji Y, et al. Beyond Flood Preparedness: Effects of Experience, Trust, and Perceived Risk on Preparation Intentions and Financial Risk-Taking in China[J]. Sustainability, 2021, 13(24): 13625.

Those points suggest that other references may be problematic as well, yet I was unable to check all of them. The authors, therefore, need to go through all their references again and re-evaluate their applicability.

2. Methodology

3.3. Descriptive statistical analysis
I believe the sections should be renamed/splitted.
I miss the Sample section, which the authors describe here.

To other points: The authors write "They come from different industries and regions, and there are certain differences in positions and working years in the enterprise, which makes the sample have strong reliability and richness."

I doubt this conclusion. It looks like a description of a typical, cross-sectional convenience sample. This is by itself not bad, but far from "strong reliability and richness"

"This paper distributed a total of 317 questionnaires in August 2021".
The grammar/syntax in this sentence is misleading (papers do not distribute questionnaires).

Additionally, the authors fail to indicate how the questionairs were distributed. Authors colleagues? Students? People who lived in a certain village? Mailing list of an association? We do not know.
Then - why only 317? What is the reason for this number? Could the authors get more? What was the expected power of this sample? Which strength of effect did the authors want to archive? There is no power analysis conducted, which is a drawback in this type of study.

Then there is a small error: in table 9 authors put the numbers of statements in the first column. This is wrong for this table (as the factors do not correspond to single items, but to item budnles, which authors describe later).

A bigger problem in the factor analysis is that the authors use exploratory analysis, whereas they should use confirmatory analysis. The items/dimensions are not something new, they are rather drawn form the theory. Therefore exploratory analysis is not needed/not recommended. The authors KNOW UPFRONT which factors are there! This entire section must be changed in my opinion.

Small, last problem. The authors report that the could not confirm 3 detailed hypotheses: one from H2, and two from H3. Yet they state that both the H2 and H3 were confirmed. I would probably accept "partial confirmation" of H2 and H3, not full confirmation.

To sum up, the paper is promising, but the methods section is slightly outdated and some references are misleading. When the authors fix it, there is a potential for the paper to be accepted.

Author Response

Response to the Reviewer 1’ comments

——Round 1

I read the revised version of the manuscript with some degree of fright, as the previous version was problematic. Yet authors managed to solve most of the problems addressed in the previous review, bringing the paper to a new level of clarity. This is highly appreciated. There are still some minor things the authors could work on, to make the paper a high impact on the world of work psychology. I will list some of the improvement suggestions below:

  1. Theoretical introduction:

The authors in the text sentence relating to the reference 9 write about "better voice behavior" of employees. I do not get this phrase. Could You please explain/rephrase?

Answer: Sorry, this is a Chinese style description. "voice behavior" means that employees can express their opinions and views more boldly under positive psychological cues, instead of being bored in their hearts, which in turn affects work efficiency and passion.

The author has changed "and have better voice behavior and performance at work[9]" here to "and have better behavior and performance in expressing their opinions at work[9]".

  1. The reference 17 should relate to the formation of work self-work concept. (The process and results of evaluation have an important impact on the formation of employees' work self-concept[17]). Yet the referenced paper is: J Gómez-Odriozola, Calvete E. Effects of a Mindfulness-based Intervention on Adolescents' Depression and Self-concept: The Moderating Role of Age[J]. Journal of Child and Family Studies, 2021, 30(2): 1501-1515.

I believe this paper does not support the argument the authors bring up.

  1. Similarly as in the previous point, I see a problem with the following references:

21, 22; Authors write: "Employees who perceive the trust of their superiors have higher responsibility norms, organizational commitment and organizational self-esteem, and have a higher degree of motivation and completion for the work assigned by their superiors[21,22].", but the references talk about farmers and flood risk, and not the manger - employee relation/trust.

Answer: The author has verified this part of the content, and has improved and revised the content and references in the paper.

Self-Evaluation, as the intrinsic motivation of employees, represents how employees evaluate themselves[16], and can have a positive and significant impact on employees' behavior, resulting in the satisfaction of leaders[17].

Employees who perceive higher-level trust have higher accountability norms, organizational commitment, and organizational self-esteem. The positive relationship between employees' social responsibility and their own behavior becomes stronger when leaders motivate frontline employees to serve customers. Furthermore, when frontline employees are satisfied with their jobs, the relationship between responsibility, self-esteem and their own behavior is strengthened[21,22].

[17] Abadi S J, Mahdavipour Z, Rezaei A, et al. The relationship between employee self-concept, brand identity, brand pride and brand citizenship behaviour and customer satisfaction[J]. International Journal of Business Excellence, 2021, 23(2): 171-187.

[21] Youn H, Kim J H. Corporate Social Responsibility and Hotel Employees' Organizational Citizenship Behavior: The Roles of Organizational Pride and Meaningfulness of Work[J]. Sustainability, 2022, 14(4): 1-18.

[22] F Velasco Vizcaíno, Martin S L, Cardenas J J, et al. Employees' attitudes toward corporate social responsibility programs: The influence of corporate frugality and polychronicity organizational capabilities[J]. Journal of Business Research, 2021, 124(C): 538-546.

  1. Methodology

3.3. Descriptive statistical analysis

I believe the sections should be renamed/splitted.

I miss the Sample section, which the authors describe here.

To other points: The authors write "They come from different industries and regions, and there are certain differences in positions and working years in the enterprise, which makes the sample have strong reliability and richness."

I doubt this conclusion. It looks like a description of a typical, cross-sectional convenience sample. This is by itself not bad, but far from "strong reliability and richness"

"This paper distributed a total of 317 questionnaires in August 2021".

The grammar/syntax in this sentence is misleading (papers do not distribute questionnaires).

Additionally, the authors fail to indicate how the questionairs were distributed. Authors colleagues? Students? People who lived in a certain village? Mailing list of an association? We do not know.

Then - why only 317? What is the reason for this number? Could the authors get more? What was the expected power of this sample? Which strength of effect did the authors want to archive? There is no power analysis conducted, which is a drawback in this type of study.

Answer: The author has added the source and description of the questionnaire and samples in "3.3. Questionnaire" here.

3.3. Questionnaire

The questionnaire designed in this paper contains four parts. The first part is the research description of the questionnaire, including the basic information of the participants in the questionnaire, which contains 6 items. The second part is the measurement of Perceived Leader Trust, which contains 9 items in total, including 4 items for Perceived Leader Dependence and 5 items for Perceived Information Disclosure. The third part is the measurement of Psychological Empowerment, which contains 12 items, including 3 items each for Work Meaning, Ability, Autonomy, and Influence. The fourth part is the measurement of Employee Work Performance, which contains 11 items, including 5 items on Employee Task Performance and 6 items on Employee Relationship Performance.

The survey subjects selected in this paper are mainly employees of different enterprises and institutions.

Firstly, this paper selected MBA students who had participated in actual work in enterprises and institutions. They came from different industries and regions, which made the sample highly reliable and rich. A total of 110 questionnaires were distributed and 101 questionnaires were returned.

Secondly, relying on Internet social platforms-"WeChat" and "QQ" to distribute questionnaires to classmates, friends, etc. participating in the work, a total of 207 questionnaires were distributed and recovered.

The questionnaire was issued from June 2021 to August 2021. A total of 317 questionnaires were distributed and 308 questionnaires were collected. The sample recovery rate was 97.2%. After removing 21 invalid questionnaires, 287 questionnaires were obtained, and the recovery rate of valid questionnaires was 90.5%.

*NOTE: All methods were carried out in accordance with relevant international and Chinese guidelines and regulations. All experimental protocols were approved by Institute of Psychology, Chinese Academy of Sciences, and Ethics Committee of CAS. Moreover, the informed consent was obtained from all subjects and their legal guardian(s).

A bigger problem in the factor analysis is that the authors use exploratory analysis, whereas they should use confirmatory analysis. The items/dimensions are not something new, they are rather drawn form the theory. Therefore exploratory analysis is not needed/not recommended. The authors KNOW UPFRONT which factors are there! This entire section must be changed in my opinion.

Answer: In order to simplify the content of the manuscript, the author has added "Appendix" at the end of the paper.

Small, last problem. The authors report that the could not confirm 3 detailed hypotheses: one from H2, and two from H3. Yet they state that both the H2 and H3 were confirmed. I would probably accept "partial confirmation" of H2 and H3, not full confirmation.

Answer: The author has changed the test results for "H2" and "H3" to "Partial Valid".

To sum up, the paper is promising, but the methods section is slightly outdated and some references are misleading. When the authors fix it, there is a potential for the paper to be accepted.

Reviewer 2 Report

The topic of the paper is aligned  to the recent findings and can provide valuable information and data.

I recommend to introduce a procedure part, before the description of the instruments used. I also recommend to use appendixes - since there are a lot of tables (the factor analysis, for example could be attached as an appendix)

Another recommendation would be the improvement of the discussion part, with direct link to the literature.

Author Response

Response to the Reviewer 2’ comments

——Round 1

The topic of the paper is aligned to the recent findings and can provide valuable information and data.

I recommend to introduce a procedure part, before the description of the instruments used. I also recommend to use appendixes - since there are a lot of tables (the factor analysis, for example could be attached as an appendix)

Answer: The author has added a description of the questionnaire composition and sample formation in the position of "3.3. Questionnaire". At the same time, in order to simplify the content of the manuscript, the author has added "Appendix" at the end of the paper.

3.3. Questionnaire

The questionnaire designed in this paper contains four parts. The first part is the research description of the questionnaire, including the basic information of the participants in the questionnaire, which contains 6 items. The second part is the measurement of Perceived Leader Trust, which contains 9 items in total, including 4 items for Perceived Leader Dependence and 5 items for Perceived Information Disclosure. The third part is the measurement of Psychological Empowerment, which contains 12 items, including 3 items each for Work Meaning, Ability, Autonomy, and Influence. The fourth part is the measurement of Employee Work Performance, which contains 11 items, including 5 items on Employee Task Performance and 6 items on Employee Relationship Performance.

The survey subjects selected in this paper are mainly employees of different enterprises and institutions.

Firstly, this paper selected MBA students who had participated in actual work in enterprises and institutions. They came from different industries and regions, which made the sample highly reliable and rich. A total of 110 questionnaires were distributed and 101 questionnaires were returned.

Secondly, relying on Internet social platforms-"WeChat" and "QQ" to distribute questionnaires to classmates, friends, etc. participating in the work, a total of 207 questionnaires were distributed and recovered.

The questionnaire was issued from June 2021 to August 2021. A total of 317 questionnaires were distributed and 308 questionnaires were collected. The sample recovery rate was 97.2%. After removing 21 invalid questionnaires, 287 questionnaires were obtained, and the recovery rate of valid questionnaires was 90.5%.

*NOTE: All methods were carried out in accordance with relevant international and Chinese guidelines and regulations. All experimental protocols were approved by Institute of Psychology, Chinese Academy of Sciences, and Ethics Committee of CAS. Moreover, the informed consent was obtained from all subjects and their legal guardian(s).

Another recommendation would be the improvement of the discussion part, with direct link to the literature.

Answer: The author has revised this part and summed up "Discussion" and "Conclusion" respectively.

  1. Discussion

(1) Perceived Leader Trust positively affects Employee Work Performance

The regression coefficients of Perceived Leader Dependence and Perceived Information Disclosure on Employee Task Performance are 0.483 and 0.617; the regression coefficients of Perceived Leader Dependence and Perceived Information Disclosure on Employee Relationship Performance are 0.428 and 0.494. This result shows that employees will more actively complete the tasks assigned by their leaders because they perceive their leaders' dependence and information disclosure, and thus have better performance at work.

(2) Perceived Leader Trust positively affects employees’ Psychological Empowerment level

The empirical results show that Perceived Leader Trust has a positive impact on the overall Psychological Empowerment of employees. Perceived Leader Dependence has a significant positive effect on the Work Meaning, Ability, Autonomy and Influence of Psychological Empowerment, and Perceived Information Disclosure has a positive impact on employees' Ability, Autonomy and Influence. Perceived trust based on dependence and information disclosure is built on the emotional connection, interpersonal interest, and support of leaders and subordinates, while employees' perception of Psychological Empowerment is closely linked to superiors' communication and support. Therefore, Perceived Leader Trust can positively affect employees' Psychological Empowerment.

(3) Psychological Empowerment positively affects Employee Work Performance

The empirical results show that employees' overall Psychological Empowerment has a positive impact on Employee Work Performance. Employees with high Psychological Empowerment tend to be proactive in their work, and have more input in their work, which in turn promotes employees to have higher Employee Work Performance. The four dimensions of Psychological Empowerment can positively affect Employee Task Performance, the Ability and Influence of Psychological Empowerment have a positive impact on Employee Relationship Performance, and Work Meaning and Autonomy have no significant impact on Employee Relationship Performance. The reason for this result is that Employee Relationship Performance is more dependent on the performance and influence of employees at work. However, employees' perception of Autonomy emphasizes the degree of employees' self-determination of work, which is not much related to Employee Relationship Performance.

(4) Psychological Empowerment plays a partial mediating role between Perceived Leader Trust and Employee Work Performance

Psychological Empowerment, as a whole, plays a partial mediating role between Perceived Leader Dependence and Employee Task Performance, and partially mediates between Perceived Leader Dependence and Employee Relationship Performance. When employees feel the trust of their leaders, their Employee Work Performance is positively affected, and the effect of Perceived Leader Trust can be explained by changes in employees' Psychological Empowerment.

  1. Conclusion

Based on the research results of previous scholars, this paper constructs a theoretical model of Perceived Leader Trust, Psychological Empowerment and Employee Work Performance, and proposes 28 research hypotheses. Among them, Perceived Leader Trust is divided into Perceived Leader Dependence and Perceived Information Disclosure; Psychological Empowerment is divided into Work Meaning, Ability, Autonomy and Influence; Employee Work Performance is divided into Employee Task Performance and Employee Relationship Performance. This paper adopts a combination of online (WeChat and QQ) and offline (MBA students) methods to collect 308 research data, verify the theoretical model and research hypothesis constructed in this paper through empirical analysis, and finally draw the research conclusion.

(1) Perceived Leader Trust has a positive impact on Employee Work Performance. (2) Perceived Leader Trust can positively affect employees' perception of Psychological Empowerment. Among them, Perceived Leader Dependence has a significant impact on all dimensions of Psychological Empowerment, but the relationship between Perceived Information Disclosure and Work Meaning is not significant. (3) Employees' Psychological Empowerment perception is positively related to their work performance. Among them, the four dimensions of Psychological Empowerment are significantly related to Employee Task Performance, and the relationship between Work Meaning and Autonomy and Employee Relationship Performance is not significant. (4) Psychological Empowerment, as the overall perception of employees, plays a partial mediating role between Perceived Leader Trust and Employee Work Performance.

Reviewer 3 Report

The manuscript deals with the relationship patterns among a group of variables implied in organizational labor perceptions. The final purpose analyzes the mediating role of empowerment between leader trust and work performance. This can be an interesting study, but I think there are several methodological issues that introduce doubts about data consistencies. These doubts make me neglect it for publication.

Specifically, the sample is an incidental and short sample. A final group, close to 300 participants answered three inventories. There is no data on how participants were recruited. Two of these inventories were developed for this research.  For that, the manuscript provides exploratory factor analyses and internal consistencies data. I think this information is insufficient to establish the construct and convergent validity. Besides, employee work performance was also measured as it was perceived by workers themselves, without any control about its social desirability.

Manuscript tries to answer 28 hypotheses. These hypotheses are redacted as general and universal sentences, despite the sample size, its representativeness, and inventories administered. Also, hypotheses use the verb “affects”. In line with this, manuscript makes assertions as generalized findings, as It is done in the abstract: “This paper verifies the role of Psychological Empowerment between Perceived Leader Trust and Employee Work Performance, and explores the internal mechanism of Perceived Leader Trust from the perspective of employees' Intrinsic Work Motivation…”. First, a cross-sectional observational design cannot establish causal relations among variables. Second, data must be taken cautiously, far from those general statements.

I can see and appreciate the great work done by author/s and the suitability of mediating data analysis. But, I am sorry, there are essential methodological issues that make every result an inconsistent finding.

Author Response

Response to the Reviewer 3’ comments

——Round 1

The manuscript deals with the relationship patterns among a group of variables implied in organizational labor perceptions. The final purpose analyzes the mediating role of empowerment between leader trust and work performance. This can be an interesting study, but I think there are several methodological issues that introduce doubts about data consistencies. These doubts make me neglect it for publication.

Specifically, the sample is an incidental and short sample. A final group, close to 300 participants answered three inventories. There is no data on how participants were recruited. Two of these inventories were developed for this research. For that, the manuscript provides exploratory factor analyses and internal consistencies data. I think this information is insufficient to establish the construct and convergent validity. Besides, employee work performance was also measured as it was perceived by workers themselves, without any control about its social desirability.

Answer: The author has added a description of the questionnaire composition and sample formation in the position of "3.3. Questionnaire". This will make the source and reliability of the data more convincing.

3.3. Questionnaire

The questionnaire designed in this paper contains four parts. The first part is the research description of the questionnaire, including the basic information of the participants in the questionnaire, which contains 6 items. The second part is the measurement of Perceived Leader Trust, which contains 9 items in total, including 4 items for Perceived Leader Dependence and 5 items for Perceived Information Disclosure. The third part is the measurement of Psychological Empowerment, which contains 12 items, including 3 items each for Work Meaning, Ability, Autonomy, and Influence. The fourth part is the measurement of Employee Work Performance, which contains 11 items, including 5 items on Employee Task Performance and 6 items on Employee Relationship Performance.

The survey subjects selected in this paper are mainly employees of different enterprises and institutions.

Firstly, this paper selected MBA students who had participated in actual work in enterprises and institutions. They came from different industries and regions, which made the sample highly reliable and rich. A total of 110 questionnaires were distributed and 101 questionnaires were returned.

Secondly, relying on Internet social platforms-"WeChat" and "QQ" to distribute questionnaires to classmates, friends, etc. participating in the work, a total of 207 questionnaires were distributed and recovered.

The questionnaire was issued from June 2021 to August 2021. A total of 317 questionnaires were distributed and 308 questionnaires were collected. The sample recovery rate was 97.2%. After removing 21 invalid questionnaires, 287 questionnaires were obtained, and the recovery rate of valid questionnaires was 90.5%.

*NOTE: All methods were carried out in accordance with relevant international and Chinese guidelines and regulations. All experimental protocols were approved by Institute of Psychology, Chinese Academy of Sciences, and Ethics Committee of CAS. Moreover, the informed consent was obtained from all subjects and their legal guardian(s).

Manuscript tries to answer 28 hypotheses. These hypotheses are redacted as general and universal sentences, despite the sample size, its representativeness, and inventories administered. Also, hypotheses use the verb “affects”. In line with this, manuscript makes assertions as generalized findings, as It is done in the abstract: “This paper verifies the role of Psychological Empowerment between Perceived Leader Trust and Employee Work Performance, and explores the internal mechanism of Perceived Leader Trust from the perspective of employees' Intrinsic Work Motivation…”. First, a cross-sectional observational design cannot establish causal relations among variables. Second, data must be taken cautiously, far from those general statements.

Answer: Trust and perceived trust as components of interpersonal trust are two independent constructs. The fact that employees perceive trust affects their behavior only when they feel they are trusted. Current research mainly relies on two mechanisms to explain the effects of Perceived Leader Trust: social exchange mechanisms based on reciprocity norms and cognitive mechanisms. These two mechanisms describe Perceived Leader Trust as a coercive or reactive role, but they do not fully capture the effects of perceived trust. In fact, Perceived Leader Trust can be more proactive and self-driven. This paper captured the nature of the new role played by Perceived Leader Trust based on the lens of Intrinsic Work Motivation and Self-Evaluation. The internal mechanism of perceived trust was explored, with Perceived Leader Trust as an independent variable, Employee Work Performance as a dependent variable, and Psychological Empowerment as a mediating variable.

I can see and appreciate the great work done by author/s and the suitability of mediating data analysis. But, I am sorry, there are essential methodological issues that make every result an inconsistent finding.

Round 2

Reviewer 3 Report

I am very sorry, but my concerns about methodological insufficiency practically remain, and, consequently, my opinion also remain.

Author Response

The author has further improved the methodology of the research, hoping to be recognized by the reviewers and editor.

This manuscript is a resubmission of an earlier submission. The following is a list of the peer review reports and author responses from that submission.

Round 1

Reviewer 1 Report

Dear Author,

thank you for inviting me to review your manuscript. After reading your manuscript I think there are some major problems that made it difficult for me to read.

The manuscript does not follow a proper structure for an article: Introduction, Objectives, Material and methods, Discussion and Conclusions. Although your manuscript appears as an article, the first part or the introduction is very broad and seems to be a theoretical review. The objective of the study does not appear, the methodology is not well explained, the discussion section does not appear and the conclusions are very long, perhaps due to the lack of objective.

The summary is not structured correctly either. The objective does not appear, nor do the results of the analyses carried out, only the conclusions of these analyses.

These problems make it difficult to read and its scientific character very low.

I advise you to rewrite your manuscript, paying special attention to the structure to be followed, and with the recommendations I have mentioned above.

Thank you very much.

Kind regards.

Reviewer 2 Report

I found the topic of the paper to be highly interesting. The authors try to show (analyse) the mediating role of psychological trust in the relation between perceived trust and work performance. But there are serious flaws mostly in the style of writing, which cast a big shadow on the readability of the theoretical part and make it impossible to properly evaluate the paper.

Unfortunately, in almost every paragraph there are a couple or more serious flaws, which render reading (and understanding) of the message impossible. I will show it on one example (lines 76-81), one of the most clear in the whole paper:

"This article explores the mechanism of perceived trust, which has contributed to the development of the theory of interpersonal trust"

I am quite confident the authors do not elaborate on the mechanism of perceived trust. They do try to split perceived trust on a couple of viewpoints, but they do not show any mechanism behind it.
I also do not understand how the postulated mechanism of perceived trust would contribute to the theory of interpersonal trust.

"the introduction of intrinsic work motivation and self-evaluation perspectives is a new supplementary method to explain the mechanism of perceived trust."

This sentence is supposed to suggest, that the authors add something new (to the topic). There are many problems with this sentence. How is the intrinsic work motivation added? As a perspective? If so, is it still a method of explanation? Then how the intrinsic motivation explains the mechanism of perceived trust? I was unable to get it clearly from the article. The same goes to self-evaluation perspective. I am not sure if intrinsic motivation and self-evaluation can be used to explain (perceived) trust, and I was not convinced otherwise by the article.

"In the study of perceived trust and work performance, this article focuses on the mediating role of psychological empowerment, and further understands the internal mechanism of perceived trust."

This first part of this sentence makes good sense, as the authors do in fact study (analyse) the relation between perceived trust and work performance and the mediating role of empowerment between them. But the second part is confusing. Who or what understands the mechanism? Maybe the authors mean explain? Even if so, what explains what? It is unclear from the sentence...

Even if the theoretical introduction would be cleared, the hypotheses would need to be rewritten, the authors would need get rid of explaining the process (like in the line180-193).

Even if they do, then the sample is badly gathered -only a convenience sample, mostly students and authors colleagues from social media. Then the analyses are quite superficial.

The final problem is a very sparse discussion of the results, which only shows what the authors got (some of the results), and does not relate the results to the theory presented at the start.

To sum up - the topic of the analysis is valid, but the approach is lacking in almost every aspect of the study, making the final article difficult to read. In the end, the article in the present form does not make a positive contribution to science and should therefore not be printed.